# Multi-Level Spatial Embedding Sharing for Enhanced Online Trajectory-User Linking

## Abstract

Trajectory-User Linking (TUL) is a critical task in mobility applications that links unlabeled spatial trajectories to the users or entities that generated them. In these applications, data often arrives as a continuous stream and may experience distributional shifts over time. While adapting TUL models via online learning could address these challenges, this approach remains unexplored in current research. Our work bridges this gap by conducting comprehensive evaluations of common TUL techniques in an online learning context. To improve the performance of existing TUL techniques in this setting, we further introduce a novel embedding approach called Multi-Level Spatial Embedding Sharing (MiLES). MiLES operates by partially sharing embeddings for locations within neighborhoods of multiple size levels. This design enables faster adaptation via frequently-updated shared embeddings, while maintaining fine-grained discrimination through more location-specific representations. MiLES also significantly reduces the number of embedding parameters leading to lower memory usage and more computationally efficient model updates. We further incorporate learnable weighting parameters for each embedding level, allowing the model to dynamically adjust the influence of different levels based on incoming data. Our experimental results on several real-world datasets show that integrating MiLES into state-of-the-art TUL models significantly improves their performance in online learning scenarios, yielding relative gains in top-1 accuracy of up to 24%. While developed for TUL, MiLES is broadly applicable to other spatial machine learning tasks benefitting from multi-scale representations. Our code is available at `https://anonymous.4open.science/r/MiLES-3D20`.

## 1 Introduction

The proliferation of location-enabled devices has produced vast amounts of mobility data, fueling advances in various machine learning tasks (Rehman et al., 2015; Zheng, 2015). One such task is Trajectory-User Linking (TUL), introduced by Gao et al. (2017). TUL aims to associate chronologically ordered sequences of check-ins at visited locations or points of interest (POIs), so-called trajectories, to the users or entities that generated them. The TUL task has numerous real-world applications, including disease control, law enforcement, ride-sharing, and location-based recommendations (Hao et al., 2020; Gao et al., 2017). Traditional approaches for TUL have demonstrated impressive performance using recurrent neural networks and, more recently, transformer-based architectures. However, these methods are designed for batch learning, where all training data is available beforehand, and to the best of our knowledge, no prior work has investigated TUL in online learning settings. In many practical scenarios, such as the deployment of a new location-based service, initial data is often limited, and data instead arrives over time as part of a continuous stream. In addition, mobility data streams are subject to temporal dependencies and distributional shifts, commonly known as concept drift, caused by factors such as new POIs, changed traffic routing, or evolving user behavior (Borkowski et al., 2021; Gomes et al., 2017). The COVID-19 pandemic, for instance, caused significant distributional shifts in mobility patterns (Borkowski et al., 2021), demonstrating the necessity for models that can adapt dynamically. Online learning offers a promising solution to these challenges by allowing models to incrementally update their knowledge with new trajectory data. In contrast to traditional batch learning, online models process data sequentially

as it arrives, enabling adaptation to changing mobility patterns. According to the definition of online machine learning by Bifet et al. (2010), a model operating in such an environment must be able to:

**R1**: process a single instance at a time,

**R2**: process each instance in a limited amount of time,

**R3**: use a limited amount of memory,

**R4**: predict at any time,

**R5**: adapt to changes in the data distribution.

Most existing TUL approaches can be adapted for online TUL, which we formalize in Equation 2, by updating models incrementally. However, their ability to predict at any time and adapt to distributional shifts can be hindered by their embedding strategy. To embed check-in locations, most models rely on lookup tables containing separate learnable embedding vectors for each POI. For any given trajectory, only the embeddings for the visited locations receive non-zero gradients with this approach. However, given the large number of unique POIs in most TUL applications, most locations receive relatively few check-ins (Chen et al., 2022). This results in the gradients for POI-based embeddings often suffering from a high degree of sparsity. While in batch-learning, this sparsity can be mitigated by training on a large collection of trajectories, it may hurt the performance in an online learning setting, where quick adaptation to new data is crucial (**R5**).

A potential solution is to share embeddings across multiple locations to reduce gradient sparsity, as was explored in previous works on batch-learning TUL (Yang et al., 2021; Alsaeed et al., 2023). Sharing embeddings based on spatial proximity allows models to generalize knowledge across locations, improving adaptation for rarely visited POIs. However, there is a trade-off: larger neighborhood sizes reduce the gradient sparsity, but also reduce the specificity of embeddings. This means that while broad embedding sharing may be beneficial at the start of training or following a concept drift, it likely degrades the model's ability to learn fine-grained location details.

To address this issue, we propose Multi-Level Spatial Embedding Sharing (MiLES), a novel embedding approach for online TUL. MiLES partitions embeddings into multiple segments, each shared across neighborhoods of different sizes. This design enables a wide spectrum of representations with varying levels of adaptation speed, while also reducing the overall parameter count and improving the computational efficiency of model updates compared to a purely POI-based approach without shared embeddings. MiLES additionally features a learnable weighting mechanism that dynamically adjusts the contribution of each embedding level, allowing the model to emphasize broader or finer-grained spatial information as needed during different stages of online learning.

To evaluate the effectiveness of MiLES, we first conduct a comprehensive assessment of existing state-of-the-art TUL techniques in an online learning setting. We then integrate MiLES into these techniques and demonstrate its significant performance improvements. Lastly, we perform ablation studies to analyze the contribution of each MiLES component and gain deeper insights into its functionality.

While our primary focus is on TUL, the modular and task-agnostic design of MiLES makes it applicable to other spatial machine learning tasks that benefit from multi-scale representations. To demonstrate this versatility, we successfully apply MiLES to destination prediction, showing its broader utility beyond TUL (see Appendix C.1).

## 2 PRELIMINARIES

In the following we will introduce the basic concepts underlying online trajectory user linking:

**Definition 2.1** (Check-In). A check-in is a tuple $c = (t, l)$ containing a timestamp $t$ and the coordinates $l$ of a visited location of form $(\text{latitude}, \text{longitude})$. Check-ins often additionally contain a unique identifier $p \in \mathbb{P}$ for the Point Of Interest (POI) at $l$, where $\mathbb{P}$ is a finite set of POI identifiers. For simplicity, we use "POI" and "location" interchangeably.

**Definition 2.2** (Trajectory). A trajectory $T$ is a chronologically ordered sequence of $n$ check-ins, $T = [c_0, c_1, \ldots, c_n]$. Each trajectory is generated by a single user $u$ from a set of users $\mathbb{U}$. A pair $(T, u)$ is a *linked trajectory*. When the generating user is unknown, the trajectory $T$ is considered *unlinked*.

**Definition 2.3** (Trajectory User Linking). The task of trajectory-user linking is to learn a function $f$ on a training set of linked trajectories $\mathbb{D} = \{(T_0, u_0), \ldots, (T_n, u_n)\}$, that correctly assigns a user label to each unlinked trajectory in a test set $\mathbb{D}^{(\text{test})}$. Formally, the objective is to minimize the predictive loss:

$$\min_{\boldsymbol{\theta}} \sum_{(T_i, u_i) \in \mathbb{D}^{(\text{test})}} L(f(T_i; \boldsymbol{\theta}, \mathbb{D}^{(\text{train})}), u_i), \tag{1}$$

where $L$ is a loss function quantifying the predictive error, and $\boldsymbol{\theta}$ are the model parameters to be optimized.

In an online setting, where samples arrive sequentially and cannot be stored indefinitely, TUL is more effectively evaluated under the *prequential* or *interleaved test-then-train* scheme (Bifet et al., 2010), where each incoming sample is first used to test the model and then for updating it. Given a stream of pairs $\mathbb{S} = \{(T_0, u_0), \ldots, (T_m, u_m)\}$, the objective becomes:

$$\min_{\boldsymbol{\theta}_0, \ldots, \boldsymbol{\theta}_{m-1}} \sum_{i=1}^{m-1} L(f(T_i; \boldsymbol{\theta}_{i-1}, \mathbb{S}_{:,:i-1}), u_i), \tag{2}$$

where $\boldsymbol{\theta}_{i-1}$ are the model parameters at time $i-1$, and $\mathbb{S}_{:,:i-1}$ denotes all previously seen pairs. Therefore, the parameters at each step of the training process contribute equally to the performance of an online TUL model, making it more susceptible to the embedding sparsity issue mentioned in section 1.

# 3 RELATED WORK

While trajectory user linking itself is a relatively recent task, it builds upon established methods from adjacent fields. Early approaches adapted the longest common subsequence (LCS) algorithm (Ying et al., 2011) to predict user labels by finding the longest shared sub-trajectory between unlabeled and known trajectories. Similarly, bag-of-words representations, which encode trajectories based on POI visit frequencies (Mikolov et al., 2013), enable the application of conventional classification methods such as linear discriminant analysis and support vector machines to the TUL problem.

More recent approaches, starting with TUL via Embedding and RNN (TULER) (Gao et al., 2017), use a lookup-table embedding scheme that preserves the temporal order of check-ins. Gao et al. (2017) introduced three TULER variants (TULER-G, TULER-L and BiTULER) combining this embedding approach with GRU (Cho et al., 2014), LSTM or bidirectional LSTM networks (Hochreiter & Schmidhuber, 1997). In subsequent works, various extensions of TULER were proposed, including TULVAE (Zhou et al., 2018) which combines an LSTM classifier with a variational autoencoder, and DeepTUL (Miao et al., 2020) which extends TULER with a historical attention module based on user IDs of previous check-ins sharing the same locations and time-slots.

With their advancement in other machine learning disciplines, newer studies on TUL have increasingly focused on transformer-based approaches. The T3S model (Yang et al., 2021) combines transformer and LSTM encoders to encode trajectories before classification. Similar to our proposed MiLES approach, T3S embeds locations by mapping them to grid cells, but uses only a single grid with fixed resolution, limiting its embeddings due to the trade-off between gradient-density and specificity. The purely transformer-based TULHOR (Alsaeed et al., 2023) embeds check-ins using hexagonal grids and supplements these with conventional POI embeddings. Another relevant use of multi-scale encoding is in time-of-arrival estimation, where Hu et al. (2022) represent locations with geohashes at varying resolutions and combine them via feature hashing. Their work demonstrates the utility of capturing spatial information at different granularities. MiLES makes use of the same concept but introduces two key distinctions: a learnable weighting mechanism that allows the model to adaptively focus on different scales, and a structured partitioning of the embedding dimensions, as detailed in Section 4 to address the challenges of online learning TUL.

Other notable TUL models have employed approaches include MainTUL (Chen et al., 2022), which combines transformers and RNNs through mutual distillation, and TGAN (Zhou et al., 2021c), which uses GANs for data augmentation. Further TUL models include GNNTUL (Zhou et al., 2021a) and AttnTUL (Chen et al., 2024), which process trajectories using graph neural networks,

the self-supervised learning approach SML (Zhou et al., 2021b), as well as the Siamese neural network TULSN (Yu et al., 2020).

Outside of TUL, Fourier features (Tancik et al., 2020) are commonly used for applications like Neural Radiance Fields (Mildenhall et al., 2020), mapping coordinates into a higher-dimensional space using multiple sinusoidal functions of varying frequencies.

Although they address different aspects of the learning process and are therefore complementary to embedding techniques, general online machine learning techniques may also improve convergence of TUL models in an online setting, including replay methods (see e.g. Mnih et al., 2015; Lillicrap et al., 2019; Prabhu et al., 2020) and adaptive optimizers like Hypergradient Descent (Baydin et al., 2018) and DoG (Ivgi et al., 2023).

## 4 MULTI-LEVEL SPATIAL EMBEDDING SHARING

Existing TUL approaches use lookup-table embeddings to encode the spatial information of check-ins. This embedding method uses a matrix $\boldsymbol{Z}$ of shape $|\mathbb{L}| \times d$, where $\mathbb{L}$ is the set of unique check-in locations. The embedding function can be denoted as $z(i) = \boldsymbol{Z}_i$, where $i$ is the index of a check-in location $\boldsymbol{l} \in \mathbb{L}$ and $\boldsymbol{Z}_i$ is the $i$-th row vector of the embedding matrix. Given that only a single row is selected for any given location $\boldsymbol{l}$, the sparsity of this approach, defined as the fraction of active parameters in the embedding matrix, is simply $1 - 1/|\mathbb{L}|$. Since mobility data commonly includes thousands of unique locations, the parameter usage and therefore also the gradients of such embeddings are generally very sparse. Existing methods, such as T3S (Yang et al., 2021) and TULHOR (Alsaeed et al., 2023), attempt to mitigate this issue by sharing embeddings between locations that belong to the same cell in a predefined spatial grid. By grouping multiple locations into shared embedding cells, the number of unique embeddings is reduced, thereby increasing parameter utilization and lowering sparsity to $1 - 1/|\mathbb{H}|$, where $|\mathbb{H}|$ is the number of cells in the grid-based partitioning of the coordinate plane.

However, this shared embedding approach creates a fundamental trade-off: while grouping locations into cells reduces gradient sparsity, it also reduces the informational content of the embeddings, as multiple distinct locations are now represented by a single vector (see Appendix B.1 for a formal analysis). Embedding techniques that use a fixed level of sharing must therefore balance adaptation speed with representation specificity. However, in an online learning setting, the importance of each factor likely depends on the stage of the data stream. For instance, following a concept drift, rapid adaptation is crucial, whereas at later stages, more detailed but slower-adapting features may be preferable.

To address this challenge in online TUL, we propose multi-level spatial embedding sharing (MiLES), which generates embedding features that span a broad range of the density-information spectrum. MiLES is implemented as a drop-in replacement for standard embedding layers, adding minimal inference overhead while reducing the overall parameter count. In the following, we will give an in-depth description of the functionality of MiLES, which is also depicted in Figure 1. Like TULHOR's embedding approach (Alsaeed et al., 2023), MiLES maps locations to a grid created as a tiling of regular hexagons. We use hexagonal grids as they represent Euclidean distances more consistently than square grids (Ke et al., 2019). Unlike TULHOR, MiLES uses multiple mappings with increasing cell sizes and therefore increasing levels of aggregation.

Accordingly, we augment check-ins with an additional index $h_l$ for each embedding-level $l \in \{0, 1, ..., l^{(\max)}\}$, which identifies the specific POI for $l = 0$ and the grid cell containing the check-in location for $l > 0$. For datasets that do not provide POI identifiers, such as GeoLife (Zheng et al., 2010), $l = 0$ corresponds to the finest-resolution grid instead. We further initialize an embedding matrix $\boldsymbol{Z}_l$ of shape $h_l^{(\max)} \times d_l$ for each level, where $d_l$ is the embedding dimension and $h_l^{(\max)}$ the maximum POI- or grid cell index. To account for the decreasing informational content with increasing levels of aggregation, we assign smaller dimensions to higher-level embeddings. We compute the individual embedding dimensions as

$$d_l = \left\lfloor \frac{d \cdot \alpha^{-l}}{\sum_{l=0}^{l^{(\max)}} \alpha^{-l}} \right\rfloor, \; \alpha > 1 \tag{3}$$

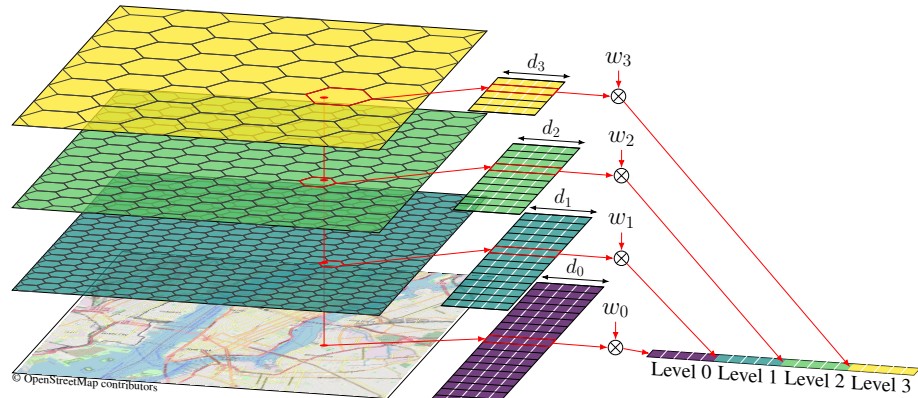

Figure 1: Visualization of the proposed multi-level spatial embedding sharing (MiLES) technique with learnable weights $w_l$ and embedding dimensions $d_l$ for each level $l \in \{0, 1, 2, 3\}$ and $\otimes$ representing scalar multiplication. Embedding dimensions are calculated according to Equation 3 but are drawn equally sized for visual clarity here.

where $d$ is the dimension of the final embedding and $\alpha$ is a hyperparameter. To reach the total number of dimensions $d$ we add the remaining dimensions to the initial embedding level. Based on the results of a hyperparameter search (see Table 5), we use $\alpha = 2$ in our experiments.

For the final embedding, we concatenate all level-specific embeddings, each weighted by a learnable parameter $w_l$. Using $\|$ to represent vector concatenation, we define the embedding function $g$ as

$$g(\boldsymbol{h}; \boldsymbol{Z}_0, \boldsymbol{Z}_1, ..., \boldsymbol{Z}_{l^{(\max)}}, \boldsymbol{w}) = \overset{l^{(\max)}}{\underset{l=0}{\Big\|}} \boldsymbol{Z}_{l,h_l} \cdot w_l, \tag{4}$$

where $\boldsymbol{Z}_{l,h_l}$ is the row vector of $\boldsymbol{Z}_l$ located at index $h_l$.

We select concatenation instead of summation to aggregate the level-specific embeddings to avoid interference between levels. For a fixed total embedding dimension, this approach also significantly reduces the number of learnable parameters, since higher-level embeddings are shared across many locations.

By multiplying the level-specific embeddings $\boldsymbol{Z}_{l,h_l}$ with learnable parameters $w_l$, we allow the average magnitude of the embedding vectors to be optimized on a per-level basis throughout the online learning process. This acts as an attention mechanism, adapting the influence of each embedding level based on the data stream. With this mechanism, the model can prioritize different embedding features based on the importance of gradient density or informational content for the current state of the data stream.

## 5 EXPERIMENTS

To evaluate the impact of the proposed MiLES approach and its individual components on the performance of existing TUL models in a data stream setting, we perform a series of experiments.

We use the widely adopted Foursquare-NYC, Foursquare-TKY (Yang et al., 2015) and GeoLife (Zheng et al., 2010) datasets. Following standard preprocessing (Chen et al., 2022), we split each trajectory into shorter segments with a maximum length of 24 hours for Foursquare-NYC and Foursquare-TKY and 3 hours for GeoLife, and selecting the most active users. For GeoLife, we additionally subsample check-ins at one-minute intervals. For more information on the selected datasets, see Table 1.

For optimization, we use Adam (Kingma & Ba, 2017). Across datasets, the input modalities include GPS coordinates, timestamps, and POI identifiers, except in GeoLife, which lacks POI information. We also provide hour-specific lookup embeddings, following prior work (see, e.g. Chen et al., 2022; Miao et al., 2020). We do not incorporate components that rely on data beyond the trajectories

Table 1: Datasets used for experimental evaluation.

| Dataset | Foursquare-NYC | | Foursquare-TKY | | GeoLife | |
|---|---|---|---|---|---|---|
| Users | 800 | 400 | 800 | 400 | 150 | 75 |
| Trajectories | 61,218 | 35,510 | 70,007 | 44,955 | 25,611 | 23,290 |
| Check-Ins | 196,435 | 137,886 | 324,564 | 248,771 | 1,284,208 | 1,187,510 |
| POIs | 34,383 | 25,443 | 38,212 | 28,286 | — | — |

themselves (e.g., TULHOR's mobility flows), as such information is absent from standard trajectory datasets and lies outside the scope of our embedding-based approach.

We tune all hyperparameters by maximizing mean prequential top-1 accuracy on the first 5,000 trajectories from the 400-user Foursquare-TKY stream. This procedure is applied consistently for all models and methods including MiLES, with tuning data excluded from evaluation. We then evaluate adaptive performance using the prequential scheme described in Section 2.

For methods requiring historical data (MainTUL and DeepTUL), we maintain a buffer of the last 1,000 trajectories. To determine embedding sharing levels, we partition the map into multiple hexagonal grids and tune the number of rows using the same protocol. The best configuration uses three levels with 200 rows at the base, halving the number of rows at each subsequent level. For GeoLife, POI embeddings are replaced with a grid-based embedding of 800 rows.

A complete list of hyperparameters is provided in Table 5. To ensure fairness, the total embedding dimensionality is fixed across methods (after baseline tuning), so that performance differences reflect only the embedding strategy rather than representational capacity.

To address the impact of MiLES' hyperparameters, we conduct a detailed analysis of the effect of the number of embedding levels and grid resolutions on model performance, as shown in Figure 2b. Standard deviations are omitted from tables for brevity. All reported results represent averages over 5 independent runs, with 90% of standard deviations below 0.29% for top-1 accuracy and below 0.27% for top-5 accuracy.

# 6 RESULTS

In the following, we will describe the results of our experimental evaluations. We evaluated various TUL approaches with their original location embedding techniques for online learning applications. The results of these experiments for the higher dataset variants with higher user counts are depicted in the upper section of Table 2 (see Table 6 for all results). As Table 2 shows, the bidirectional-LSTM-based BiTULER achieved the overall best performance on all datasets, except for GeoLife where DeepTUL yielded a higher top-1 accuracy and macro F1 score. This exception likely stems from BiTULER being the most lightweight model with the fewest parameters, enabling faster adaptation. Similarly, the generally lower performance of the transformer-based models (MainTUL, T3S, and TULHOR) compared to their RNN-based counterparts may reflect the challenges of adapting more complex architectures in online learning scenarios.

We then replaced the original embedding modules in each TUL model with MiLES, keeping the total embedding dimension constant. The bottom section of Table 2 shows the relative change in performance metrics measured in percentage points achieved by using MiLES.

The most substantial improvements were on GeoLife, where MiLES increased top-1 accuracy by up to 8.72 percentage points (a 23% relative gain) and top-5 by up to 6.98. This larger gain for GeoLife likely stems from its lack of POI information, where MiLES' additional embedding levels provide particular value compared to existing techniques' original embedding approach. Notably, improvements were consistent across all models, including T3S and TULHOR, which already incorporated single-level grid-based embeddings, demonstrating the significant benefits of MiLES' multi-level approach in online learning settings. Furthermore, since the embedding dimensionality was tuned only for the default embedding approach, the reported gains are conservative and could likely be improved with MiLES-specific tuning.

Table 2: Top-1 accuracy, top-5 accuracy and macro F1 score [%] averaged over prequential evaluation runs for TUL models with their original embedding technique (Default) and performance gains when using our technique instead ($\Delta$ MiLES).

| Dataset | Foursquare-NYC | | | Foursquare-TKY | | | GeoLife | | |
|---|---|---|---|---|---|---|---|---|---|
| Model | Acc@1 | Acc@5 | F1 | Acc@1 | Acc@5 | F1 | Acc@1 | Acc@5 | F1 |
| Default | | | | | | | | | |
| BiTULER | **60.12** | **67.20** | **57.83** | **63.16** | **74.91** | **61.06** | **37.56** | 70.85 | 26.69 |
| TULVAE | 59.79 | 66.77 | 57.32 | 55.82 | 66.23 | 51.68 | 37.08 | 70.45 | 25.25 |
| DeepTUL | 58.72 | 65.48 | 56.60 | 61.17 | 72.49 | 59.10 | 36.32 | **72.64** | **29.82** |
| MainTUL | 55.67 | 62.61 | 53.01 | 59.53 | 71.89 | 57.09 | 34.00 | 70.26 | 21.76 |
| T3S | 52.98 | 60.28 | 49.50 | 56.41 | 69.26 | 53.24 | 35.25 | 71.11 | 21.52 |
| TULHOR | 53.85 | 61.13 | 50.40 | 56.61 | 69.61 | 53.45 | 34.65 | 72.46 | 24.92 |
| $\Delta$ MiLES | | | | | | | | | |
| BiTULER | +1.49 | +3.58 | +1.71 | +1.40 | +2.85 | +1.30 | +8.72 | +6.98 | +6.26 |
| TULVAE | +1.79 | +3.73 | +2.04 | +3.07 | +4.43 | +3.34 | +8.01 | +6.57 | +4.74 |
| DeepTUL | +1.06 | +3.04 | +1.26 | +0.84 | +2.37 | +0.77 | +8.59 | +6.31 | +5.97 |
| MainTUL | +1.44 | +4.33 | +1.52 | +1.62 | +3.45 | +1.45 | +8.19 | +6.18 | +5.50 |
| T3S | +1.71 | +3.13 | +2.10 | +1.70 | +2.79 | +1.91 | +6.98 | +4.80 | +5.18 |
| TULHOR | +1.71 | +3.16 | +2.11 | +1.85 | +2.80 | +2.03 | +8.20 | +5.03 | +5.68 |

We further evaluated our approach against alternative embedding-, experience replay- and adaptive optimization methods. The results, averaged across all models from Table 2, are shown in Table 3. Among the embedding techniques, MiLES achieved the best overall performance across all datasets and metrics. It consistently outperformed the baseline POI-based lookup embeddings, as well as the hybrid linear and Fourier embeddings. These two alternatives combined POI-based embeddings with either a learnable linear projection or a Fourier feature encoding (Tancik et al., 2020) of the location coordinates, replacing the higher-level embeddings used in MiLES. Notably, both hybrid approaches underperformed compared to the simpler POI-only embeddings, suggesting that basic coordinate projections are less effective than fully dedicating the embedding space to POI-specific representations. For memory replay strategies, we compared a FIFO buffer and a random class-balanced buffer, each limited to 1,000 past trajectories, with one sample replayed per training step.

Table 3: Top-1 accuracy, top-5 accuracy and macro F1 score [%] of different embedding-, experience replay- and adaptive optimization methods, averaged across all models shown in Table 2. See Table 7 for the full results. The default configuration uses POI-based lookup embeddings and the Adam optimizer, without replay. Individual methods replace the default components. Linear and Fourier embeddings combine POI lookups with a linear or Fourier projection of coordinates, each contributing half the embedding dimensions.

| Dataset | Foursquare-NYC | | | Foursquare-TKY | | | GeoLife | | |
|---|---|---|---|---|---|---|---|---|---|
| Method | Acc@1 | Acc@5 | F1 | Acc@1 | Acc@5 | F1 | Acc@1 | Acc@5 | F1 |
| Default | 56.85 | 63.91 | 54.11 | 58.78 | 70.73 | 55.94 | 35.81 | 71.30 | 25.00 |
| Embedding | | | | | | | | | |
| Linear | 50.74 | 59.21 | 47.51 | 53.48 | 66.25 | 50.16 | 35.02 | 70.62 | 22.66 |
| Fourier | 45.31 | 57.79 | 41.68 | 51.18 | 67.72 | 47.49 | 34.54 | 70.56 | 23.26 |
| MiLES | **58.39** | **67.41** | **55.90** | **58.33** | **71.68** | **55.40** | **43.93** | **77.27** | **30.55** |
| Replay | | | | | | | | | |
| FIFO | 57.38 | 64.53 | 55.24 | 57.66 | 69.43 | 55.04 | 37.33 | 70.87 | 25.82 |
| Balanced | 57.19 | 64.26 | 54.72 | 56.96 | 68.66 | 53.95 | 36.40 | 68.37 | 25.36 |
| FIFO+MiLES | **58.94** | **67.88** | **56.87** | **60.53** | **73.85** | **57.74** | **45.86** | **77.49** | **31.87** |
| Optimizer | | | | | | | | | |
| DoG | 36.79 | 42.57 | 33.82 | 35.95 | 45.74 | 34.00 | 29.12 | 60.59 | 21.14 |
| AdamHD | 45.94 | 51.80 | 43.18 | 40.15 | 49.03 | 37.89 | 22.24 | 48.18 | 17.09 |

Table 4: Average wall time in milliseconds per test-then-train iteration for BiTULER using MiLES or its original embedding technique. Measured on a system with an Intel i5-9600K CPU and an Nvidia RTX 3090 GPU.

| Dataset | Foursquare-NYC | | Foursquare-TKY | | GeoLife | |
|---|---|---|---|---|---|---|
| Embedding | Default | MiLES | Default | MiLES | Default | MiLES |
| Wall time [ms] | 7.99 | **7.38** | 8.50 | **7.81** | **11.27** | 11.66 |

Both replay strategies improved performance over the default configuration with the models original embeddings and without replay. When combining MiLES with FIFO replay (FIFO + MiLES), we observed further gains across all datasets and metrics, demonstrating that MiLES remains effective when paired with replay techniques. The adaptive optimizers DoG and AdamHD performed significantly worse than the standard Adam optimizer, used for all other methods. We attribute this to the high gradient variance introduced by training on individual samples instead of mini-batches, which destabilizes the learning rate adaptation in both methods.

Additionally, we analyzed MiLES' computational efficiency with the BiTULER model. As shown in Table 4, MiLES reduces the processing time per trajectory on the Foursquare datasets. The parameter-sharing mechanism substantially decreases the total number of embedding parameters (e.g., from approximately 54M to 39M on Foursquare-NYC), making model updates during online learning more efficient and outweighing the small inference overhead. On the GeoLife dataset, however, the baseline embedding uses a smaller grid-based table compared to the POI-based tables in other datasets, resulting in a minor runtime increase that could potentially be mitigated through implementation optimizations.

To assess the contribution of individual components in MiLES, we conducted ablation studies by systematically removing embedding levels (-L1, -L2, -L3), the learnable weighting parameters $w_l$ (-WL), and the level-dependent embedding dimensions (-VD). When excluding an embedding level, we adjusted embedding dimensions by omitting the affected level from Equation 3. In the -VD setting, we distributed the total embedding size equally among levels.

The results in Figure 2a show that embedding levels two and three each contribute noticeable and consistent gains across all datasets and metrics. While the improvements from individual levels is moderate in isolation, their cumulative effect provides a clear advantage over standard embeddings. Level one, though less impactful for top-1 accuracy, supports macro F1 performance at negligible computational cost, confirming that all levels provide meaningful contributions. Removing level

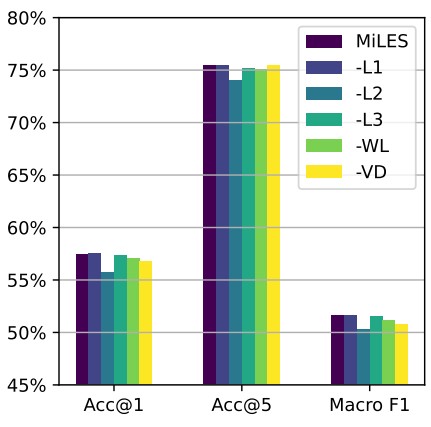

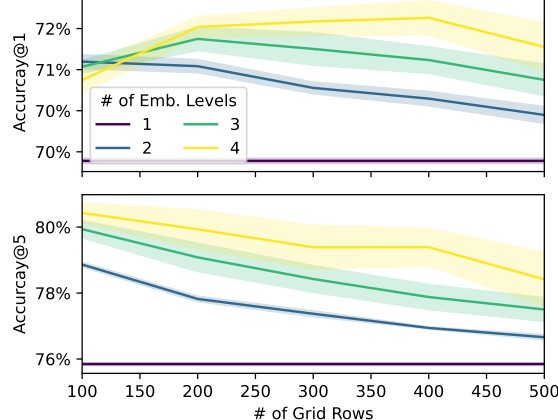

(a) Ablation study of MiLES: removing embedding levels (-L1, -L2, -L3), level weighting (-WL), or variable embedding dimensions (-VD), using BiTULER. Results are averaged over all datasets. See Table 8 for full results.

(b) Performance on the 400 user Foursquare-NYC dataset, relative to number of embedding levels and rows in the base-level grid using BiTULER and MiLES with fixed attention weights $w_l$. Shaded areas represent the $1\sigma$ range.

weighting (-WL) decreases all performance metrics, highlighting its effectiveness. The model without varying embedding dimensions (-VD) shows larger losses in top-1 accuracy and macro F1 score, likely because the uniform dimension allocation provides insufficient capacity for the more complex, fine-grained POI-level embeddings.

In another experiment, we evaluated MiLES for varying embedding aggregation levels and varying grid resolutions with the level-specific weights $w_l$ fixed at 1. For this, we used the Foursquare-NYC dataset with 400 users and BiTULER as the underlying classifier. Following previous experiments, level two used grids at half the base resolution and level three at quarter resolution. The results in Figure 2b indicate that increasing the number of embedding levels improves performance, provided that the base-level grid has a sufficiently high resolution. In particular, the four-level embedding module achieves the highest top-1 and top-5 accuracy when the base-level grid contains at least 200 rows. This suggests that MiLES is robust to variations in base grid resolution and that all embedding levels contribute meaningfully to performance. The effect of base grid resolution differs between top-1 and top-5 accuracy. While top-1 accuracy peaks at 400 grid rows for the four-level configuration, top-5 accuracy decreases steadily as resolution increases. This supports the idea that sharing embeddings across a broader set of locations benefits top-5 accuracy, as the model relies on less specific but faster adapting embeddings to correctly predict the user group.

To analyze the dynamics of the learnable level-attention weights $w_l$, we tracked their values across multiple prequential evaluation runs on the 400-user Foursquare-NYC dataset. Figure 3 shows the evolution of these weights alongside the number of unique users in the 1,000 most recent trajectories.

On average, all level-specific weights increase throughout training, effectively increasing the learning rate of the embedding parameters, as explained in Section 4. However, there are differences in how individual weights evolve. Initially, the weights associated with lower gradient sparsity embedding levels ($w_2$ and $w_3$) increase slightly faster than the POI-based embedding weight ($w_0$). After 5,000 trajectories, $w_0$ surpasses $w_1$ and $w_2$, suggesting that more specific embeddings become beneficial at this stage of online learning. However, after about 15,000 samples, both $w_0$ and $w_1$ begin to decline. This likely corresponds to an earlier concept drift, characterized by a reduction in unique users, making the coarser embeddings ($w_2$ and $w_3$) sufficient for user selection.

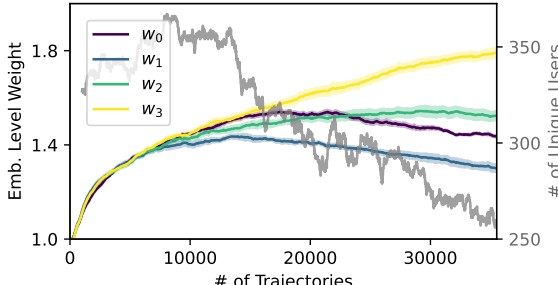

Figure 3: Mean weighting factors $w_0, ..., w_3$ and number of unique users within the last 1,000 trajectories for Foursquare-NYC. Shaded areas represent the $1\sigma$ range.

## 7    CONCLUSION

In this paper, we investigated TUL, in an online learning setting where data arrives as a stream. We introduced Multi-Level Spatial Embedding Sharing (MiLES), a novel embedding method that improves the performance of existing models in this challenging setting. MiLES balances fast adaptation and representational specificity by sharing embedding components across multiple spatial scales, using learnable weights to dynamically prioritize each scale. Our experiments demonstrate that MiLES significantly improves both the predictive accuracy and computational efficiency of state-of-the-art TUL models. As a modular component, MiLES is broadly applicable to other spatial machine learning tasks facing similar challenges of data sparsity and concept drift in streaming environments.

## USE OF LARGE LANGUAGE MODELS

Large language models were used to assist with the writing of this paper. The main functions included rephrasing sentences to improve clarity and flow, correcting grammar and suggesting more concise language. We reviewed all suggestions provided by the LLM and adapted them where necessary, ensuring that the final text accurately reflects our thoughts and intentions. We take full responsibility for all content, including the final wording and any claims or errors.

## REPRODUCIBILITY STATEMENT

Our source code, including the implementation of MiLES, dataset preprocessing and the full experimental framework, is available at `https://anonymous.4open.science/r/MiLES-3D20`. We describe the proposed approach in Section 4 and provided details regarding our experimental setup in Section 5 All experiments were conducted using publicly available datasets, which can be obtained from `https://sites.google.com/site/yangdingqi/home/foursquare-dataset` and `https://www.microsoft.com/en-us/download/details.aspx?id=52367` respectively. For an overview of our hyperparameter selection, please see Table 5.

## ETHICS STATEMENT

We acknowledge the significant privacy implications of Trajectory-User Linking. While we restricted our experiments to anonymized public datasets, the models evaluated in this paper may be used to process sensitive location data, and the online learning setting could introduce different risks to those of batch processing. While our primary focus is on predictive performance, we recognize that any real-world deployment would require robust privacy-preserving mechanisms, such as those based on differential privacy (see e.g. Mir et al., 2013). We elaborate on these ethical considerations and related work in Appendix C.3.

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

## A    EXPERIMENTAL DETAILS AND REPRODUCIBILITY

### A.1    HYPERPARAMETER VALUES

As described in the main paper, all models and techniques were tuned using the first 5,000 trajectories of the 400-user Foursquare-TKY stream. This protocol was applied identically to all methods to ensure a fair comparison. Table 5 lists the resulting hyperparameter values used in our experiments. For MiLES, $l^{(\max)} = 4$ corresponds to one POI-level embedding and three grid-based sharing levels on Foursquare datasets, and four grid-based levels on GeoLife, which lacks POI information.

Table 5: Tested hyperparameter values for the evaluated models and methods. The selected values are underlined. If different values were selected depending on the used model, multiple are underlined.

| Hyperparameter | Symbol | Tested Values |
|---|---|---|
| Full embedding dim. | $d$ | $\{512, \underline{1024}\}$ |
| Embedding levels | $l^{(\max)}$ | $\{3, \underline{4}, 5\}$ |
| Base level grid rows | $h_1^{(\max)}$ | $\{100, \underline{200}, \ldots, 500\}$ |
| Level $l$ grid rows | $h_l^{(\max)}$ | $\{\underline{2}, 3, 4\}^{-l} \times h_1^{(\max)}$ |
| Embedding dim. decay | $\alpha$ | $\{1, \underline{2}\}$ |
| Fourier feature scale | — | $\{1, 2, \underline{4} \ldots, 16\} \times 10^3$ |
| Replay samples/update | — | $\{\underline{1}, 2, 4\}$ |
| Optimizer | — | Adam |
| Learning rate | — | $\{.5, \underline{1}, \underline{2}, \underline{4}, 8\} \times 10^{-4}$ |
| Hidden layers | — | $\{\underline{1}, \underline{2}\}$ |
| Hidden units | — | $\{512, \underline{1024}\}$ |

### A.2    COMPUTING INFRASTRUCTURE

- GPU: Nvidia RTX 3090
- CPU: Intel Core i5-9600K
- System Memory: 32 GB DDR4@2133MT/s
- OS: Ubuntu 22.04 LTS
- CUDA: 12.5
- Python: 3.12
- PyTorch: 2.3.1

## A.3 FULL RESULTS FOR ALL DATASET VARIANTS

The results for all evaluated TUL approaches as well as the dataset-variants with lower user counts are displayed in Table 6. As expected, all models achieved better performance on the lower user-count datasets. In terms of individual models, BiTULER remains the overall most performant model independent of the user count, except for the GeoLife dataset where the GRU-variant of TULER and DeepTUL yielded better results.

Table 6: Top-1 accuracy, top-5 accuracy and macro F1 [%], as well as their diffences when using MiLES instead of the default embeddings, for all TUL models on all dataset variants.

| | Dataset | Foursquare-NYC | | | Foursquare-TKY | | | GeoLife | | |
|---|---|---|---|---|---|---|---|---|---|---|
| # Users | Model | Acc@1 | Acc@5 | F1 | Acc@1 | Acc@5 | F1 | Acc@1 | Acc@5 | F1 |
| | | | | | Default | | | | | |
| Low | BiTULER | **69.95** | **75.85** | **68.59** | **73.29** | **84.06** | **72.03** | **39.98** | 74.16 | 32.95 |
| | TULVAE | 69.77 | 75.72 | 68.37 | 66.86 | 77.42 | 64.05 | 39.40 | 73.82 | 31.57 |
| | DeepTUL | 68.93 | 74.77 | 67.62 | 71.65 | 82.47 | 70.42 | 38.44 | 75.15 | **33.42** |
| | MainTUL | 66.28 | 72.41 | 64.86 | 70.69 | 82.30 | 69.28 | 35.89 | 73.02 | 27.87 |
| | T3S | 63.84 | 70.42 | 61.84 | 66.77 | 79.41 | 64.89 | 37.79 | 74.68 | 29.06 |
| | TULHOR | 64.60 | 71.08 | 62.69 | 67.21 | 79.57 | 65.36 | 37.22 | **75.43** | 30.87 |
| High | BiTULER | **60.12** | **67.20** | **57.83** | **63.16** | **74.91** | **61.06** | **37.56** | 70.85 | 26.69 |
| | TULVAE | 59.79 | 66.77 | 57.32 | 55.82 | 66.23 | 51.68 | 37.08 | 70.45 | 25.25 |
| | DeepTUL | 58.72 | 65.48 | 56.60 | 61.17 | 72.49 | 59.10 | 36.32 | **72.64** | **29.82** |
| | MainTUL | 55.67 | 62.61 | 53.01 | 59.53 | 71.89 | 57.09 | 34.00 | 70.26 | 21.76 |
| | T3S | 52.98 | 60.28 | 49.50 | 56.41 | 69.26 | 53.24 | 35.25 | 71.11 | 21.52 |
| | TULHOR | 53.85 | 61.13 | 50.40 | 56.61 | 69.61 | 53.45 | 34.65 | 72.46 | 24.92 |
| | | | | | Δ MiLES | | | | | |
| Low | BiTULER | +2.15 | +3.99 | +2.43 | +1.38 | +2.24 | +1.44 | +8.97 | +6.80 | +8.40 |
| | TULVAE | +2.22 | +4.02 | +2.51 | +2.88 | +3.80 | +3.40 | +8.96 | +6.82 | +8.70 |
| | DeepTUL | +1.67 | +3.54 | +1.98 | +0.80 | +1.95 | +0.76 | +8.93 | +6.15 | +8.55 |
| | MainTUL | +2.17 | +4.67 | +2.29 | +1.25 | +2.54 | +1.08 | +9.11 | +6.65 | +8.68 |
| | T3S | +1.83 | +3.09 | +2.07 | +1.77 | +2.20 | +1.85 | +7.09 | +4.39 | +7.00 |
| | TULHOR | +1.81 | +3.00 | +2.06 | +1.64 | +2.10 | +1.65 | +8.21 | +4.71 | +7.51 |
| High | BiTULER | +1.49 | +3.58 | +1.71 | +1.40 | +2.85 | +1.30 | +8.72 | +6.98 | +6.26 |
| | TULVAE | +1.79 | +3.73 | +2.04 | +3.07 | +4.43 | +3.34 | +8.01 | +6.57 | +4.74 |
| | DeepTUL | +1.06 | +3.04 | +1.26 | +0.84 | +2.37 | +0.77 | +8.59 | +6.31 | +5.97 |
| | MainTUL | +1.44 | +4.33 | +1.52 | +1.62 | +3.45 | +1.45 | +8.19 | +6.18 | +5.50 |
| | T3S | +1.71 | +3.13 | +2.10 | +1.70 | +2.79 | +1.91 | +6.98 | +4.80 | +5.18 |
| | TULHOR | +1.71 | +3.16 | +2.11 | +1.85 | +2.80 | +2.03 | +8.20 | +5.03 | +5.68 |

The bottom section of Table 6 shows significant performance gains, when substituting the original embedding techniques of the evaluated models with MiLES for both high- and low user-count datasets. The performance benefits at lower user-counts are even higher compared to the larger dataset variants with top-1 accuracy gains of up to 9.79%. The larger benefit for data with fewer users likely stems from the fact that the lower user-count causes the individual users to be more easily identified based on the higher-level embeddings of MiLES.

## A.4 FULL RESULTS FOR EMBEDDING & REPLAY METHODS

The full results comparing MiLES to other embedding techniques as well as general online learning approaches are shown in Table 7. MiLES outperforms all other embedding techniques across all models and datasets. In many cases it even exceeds the performance of replay techniques despite that fact that MiLES adds minimal computational overhead, or reduces overhead, whereas the latter require significantly more computation. It can also be seen that MiLES can be paired with replay techniques to combine their performance benefits. For the more sophisticated TUL models (Main-TUL, T3S, TULHOR and TULVAE), a combination of FIFO and MiLES yield the best performance across all metrics and datasets.

## A.5 FULL RESULTS OF MiLES ABLATION STUDY

Table 8 presents the results of our ablation study across all evaluated datasets. The -VD variant, which uses a uniform embedding dimension for each level, performed well in terms of top-5 accuracy but was less competitive in top-1 accuracy and F1 score. The -L1 variant showed performance

Table 7: Top-1 accuracy, top-5 accuracy and macro F1 score for embedding and general online learning techniques for all TUL approaches.

| Dataset | Foursquare-NYC | | | Foursquare-TKY | | | GeoLife | | |
|---|---|---|---|---|---|---|---|---|---|
| Method | Acc@1 | Acc@5 | Macro F1 | Acc@1 | Acc@5 | Macro F1 | Acc@1 | Acc@5 | Macro F1 |
| BiTULER | | | | | | | | | |
| Lookup | 60.12 | 67.20 | 57.83 | 63.16 | 74.91 | 61.06 | 37.56 | 70.85 | 26.69 |
| Linear | 55.75 | 64.45 | 53.08 | 60.63 | 73.45 | 58.20 | 37.28 | 71.13 | 23.33 |
| Fourier | 49.28 | 61.75 | 46.07 | 55.61 | 72.33 | 52.64 | 37.35 | 71.39 | 24.92 |
| MiLES | **61.61** | **70.78** | 59.54 | **64.56** | **77.76** | **62.37** | 46.28 | **77.83** | **32.96** |
| Balanced | 60.04 | 66.95 | 57.95 | 63.08 | 74.56 | 60.89 | 36.91 | 67.83 | 25.28 |
| FIFO | 59.89 | 66.93 | 57.93 | 62.99 | 74.55 | 60.87 | 36.74 | 68.50 | 25.73 |
| FIFO + MiLES | **61.61** | 70.68 | **59.76** | 64.39 | 77.38 | 62.17 | 45.62 | 76.32 | 32.05 |
| DeepTUL | | | | | | | | | |
| Lookup | 58.72 | 65.48 | 56.60 | 61.17 | 72.49 | 59.10 | 36.32 | 72.64 | 29.82 |
| Linear | 55.86 | 64.21 | 53.58 | 59.75 | 72.12 | 57.45 | 35.74 | 72.44 | 26.61 |
| Fourier | 39.51 | 53.45 | 36.29 | 47.04 | 65.62 | 44.07 | 33.79 | 71.23 | 27.38 |
| MiLES | 59.77 | 68.52 | 57.86 | 62.01 | **74.86** | 59.87 | 44.92 | **78.95** | 35.79 |
| Balanced | 59.09 | 65.82 | 57.14 | 61.34 | 72.49 | 59.18 | 34.60 | 65.42 | 26.94 |
| FIFO | 58.86 | 65.65 | 56.91 | 61.20 | 72.47 | 59.09 | 37.89 | 71.62 | 29.75 |
| FIFO + MiLES | **60.00** | **68.70** | **58.16** | **62.21** | 74.77 | **60.02** | **47.01** | 78.53 | **36.09** |
| MainTUL | | | | | | | | | |
| Lookup | 55.67 | 62.61 | 53.01 | 59.53 | 71.89 | 57.09 | 34.00 | 70.26 | 21.76 |
| Linear | 51.48 | 60.34 | 48.19 | 57.59 | 71.04 | 54.67 | 33.48 | 69.98 | 20.25 |
| Fourier | 43.34 | 56.85 | 39.35 | 52.48 | 69.98 | 48.78 | 33.03 | 69.76 | 20.57 |
| MiLES | 57.11 | 66.94 | 54.54 | 61.15 | 75.34 | 58.54 | 42.19 | 76.44 | 27.27 |
| Balanced | 56.34 | 63.55 | 54.26 | 59.61 | 72.02 | 57.06 | 33.01 | 64.27 | 20.32 |
| FIFO | 56.11 | 63.38 | 54.14 | 59.63 | 72.15 | 57.47 | 35.81 | 69.90 | 22.08 |
| FIFO + MiLES | **57.71** | **67.65** | **55.64** | **61.30** | **75.51** | **58.92** | **44.59** | **76.87** | **28.60** |
| T3S | | | | | | | | | |
| Lookup | 52.98 | 60.28 | 49.50 | 56.41 | 69.26 | 53.24 | 35.25 | 71.11 | 21.52 |
| Linear | 39.56 | 47.52 | 35.49 | 38.65 | 52.21 | 34.85 | 33.53 | 69.51 | 19.97 |
| Fourier | 44.68 | 56.18 | 40.87 | 50.81 | 66.98 | 47.31 | 33.79 | 69.82 | 20.59 |
| MiLES | 54.69 | 63.42 | 51.60 | 58.10 | 72.05 | 55.16 | 42.23 | 75.90 | 26.70 |
| Balanced | 54.14 | 61.37 | 51.02 | 56.74 | 69.25 | 53.64 | 39.04 | 72.90 | 25.87 |
| FIFO | 54.71 | 62.06 | 52.26 | 57.43 | 69.85 | 54.94 | 37.48 | 71.68 | 23.38 |
| FIFO + MiLES | **56.10** | **64.67** | **53.77** | **59.05** | **72.30** | **56.54** | **45.12** | **77.23** | 29.01 |
| TULHOR | | | | | | | | | |
| Lookup | 53.85 | 61.13 | 50.40 | 56.61 | 69.61 | 53.45 | 34.65 | 72.46 | 24.92 |
| Linear | 46.90 | 55.26 | 42.85 | 51.77 | 65.17 | 48.29 | 32.53 | 70.22 | 22.22 |
| Fourier | 46.53 | 57.69 | 42.75 | 51.61 | 67.46 | 48.17 | 32.79 | 70.51 | 22.69 |
| MiLES | 55.56 | 64.29 | 52.51 | 58.46 | 72.41 | 55.48 | 42.85 | 77.49 | 30.61 |
| Balanced | 54.41 | 61.71 | 51.15 | 56.98 | 69.58 | 53.77 | 38.36 | 73.92 | 28.41 |
| FIFO | 55.02 | 62.40 | 52.48 | 57.75 | 70.24 | 55.13 | 36.90 | 72.33 | 26.82 |
| FIFO + MiLES | **56.44** | **65.00** | **54.01** | **59.32** | **72.64** | **56.75** | **44.65** | **77.77** | **32.21** |
| TULVAE | | | | | | | | | |
| Lookup | 59.79 | 66.77 | 57.32 | 55.82 | 66.23 | 51.68 | 37.08 | 70.45 | 25.25 |
| Linear | 54.90 | 63.46 | 51.88 | 52.48 | 63.52 | 47.53 | 37.57 | 70.43 | 23.56 |
| Fourier | 48.54 | 60.81 | 44.75 | 49.54 | 63.96 | 43.99 | 36.51 | 70.65 | 23.40 |
| MiLES | 61.57 | 70.50 | 59.36 | 58.89 | 70.66 | 55.01 | 45.10 | 77.02 | 29.99 |
| Balanced | 59.09 | 66.13 | 56.79 | 54.73 | 64.95 | 50.30 | 36.49 | 65.89 | 25.33 |
| FIFO | 59.69 | 66.79 | 57.72 | 57.71 | 68.31 | 54.35 | 39.14 | 71.21 | 27.18 |
| FIFO + MiLES | **61.78** | **70.60** | **59.86** | **60.75** | **72.56** | **57.67** | **48.17** | **78.25** | **33.29** |

similar to MiLES on Foursquare-NYC and even outperformed it on Foursquare-TKY. However, MiLES yielded noticeably better top-1 accuracy and F1 scores on GeoLife. On average, MiLES achieved the best overall performance in terms of top-1 accuracy and F1 score, although by a slight margin. It is worth noting that this margin would likely increase if another dataset with characteristics similar to GeoLife were included in the evaluation.

## B IN-DEPTH ANALYSIS OF MiLES

### B.1 FORMAL ANALYSIS OF THE DENSITY-INFORMATION TRADE-OFF

In the main paper, we argue that sharing embeddings creates a trade-off: it increases gradient density and thereby adaptation speed at the cost of informational content. This section provides a formal

Table 8: Full results of ablation study.

| Dataset | Foursquare-NYC | | | Foursquare-TKY | | | GeoLife | | |
| Method | Acc@1 | Acc@5 | F1 | Acc@1 | Acc@5 | F1 | Acc@1 | Acc@5 | F1 |
|---|---|---|---|---|---|---|---|---|---|
| MiLES | 63.92 | 72.25 | 61.96 | 66.64 | 79.53 | 64.58 | **45.88** | 79.10 | **35.82** |
| -L1 | **63.96** | 72.07 | **62.00** | **66.82** | 79.67 | **64.78** | 45.65 | **79.14** | 35.51 |
| -L2 | 63.85 | 71.84 | 61.87 | 66.58 | 79.28 | 64.55 | 39.60 | 75.18 | 30.64 |
| -L3 | 63.67 | 71.54 | 61.66 | 66.44 | 79.07 | 64.40 | 45.54 | 79.03 | 35.75 |
| -VD | 62.57 | **72.77** | 60.56 | 65.90 | **79.91** | 63.77 | 45.84 | 79.09 | 35.53 |
| -WL | 63.21 | 71.58 | 61.14 | 66.05 | 79.00 | 63.91 | 45.72 | 78.99 | 35.73 |

analysis of this trade-off by quantifying the information loss that occurs when spatial embeddings are shared across multiple locations.

The fundamental principle underlying this analysis is that when distinct locations are mapped to the same embedding, the model loses the ability to distinguish between them. This reduction in discriminability can be quantified as a decrease in Shannon entropy (Shannon, 1948).

Consider a subset $\mathbb{J}$ of location indices that correspond to at least two distinct locations within the same grid cell. We analyze two scenarios: distinct embeddings versus shared embeddings for these locations.

**Distinct Embeddings.** When each location index $k \in \mathbb{J}$ has its own unique embedding $z(k)$, the contribution of these embeddings to the total information content is measured by their entropy:

$$H_{\text{distinct}} = -\sum_{k \in \mathbb{J}} p(z(k)) \log p(z(k)). \tag{5}$$

**Shared Embeddings.** When all location indices in $\mathbb{J}$ use a single shared embedding $\mathbf{z}^{(\text{share})}$, the probability mass concentrates on this single embedding. The probability of observing $\mathbf{z}^{(\text{share})}$ becomes the sum of the individual probabilities: $p(\mathbf{z}^{(\text{share})}) = \sum_{k \in \mathbb{J}} p(z(k))$. Consequently, the entropy contribution becomes:

$$H_{\text{shared}} = -\sum_{k \in \mathbb{J}} p(z(k)) \log \left( \sum_{k \in \mathbb{J}} p(z(k)) \right). \tag{6}$$

**Information Loss Quantification.** The information loss due to sharing embeddings is given by the entropy difference:

$$H_{\text{shared}} - H_{\text{distinct}} = -\sum_{k \in \mathbb{J}} p(z(k)) \log \left( \frac{\sum_{k \in \mathbb{J}} p(z(k))}{p(z(k))} \right). \tag{7}$$

Since $p(z(k)) > 0$ for all $k \in \mathbb{J}$ and the sum $\sum_{k \in \mathbb{J}} p(z(k)) > p(z(k))$ for any individual term, it follows that

$$\log \left( \frac{\sum_{k \in \mathbb{J}} p(z(k))}{p(z(k))} \right) > 0.$$

As a result $H_{\text{shared}} < H_{\text{distinct}}$.

This entropy difference confirms that sharing embeddings reduces information content. Furthermore, as the number of locations in $\mathbb{J}$ increases, the sum $\sum_{k \in \mathbb{J}} p(z(k))$ grows larger, making the entropy difference more negative and indicating greater information loss when more embeddings are aggregated within shared cells.

### B.2 Ablation of Individual Embedding Levels

To isolate the contribution of different levels of spatial granularity, we evaluated BiTULER models trained with single embedding levels from MiLES. Figure 4 shows the top-5 accuracy over the first 10,000 trajectories.

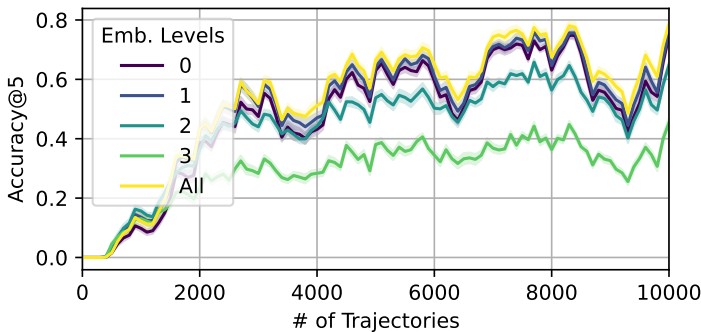

Figure 4: Top-5 accuracy of BiTULER models using individual MiLES embedding levels on Foursquare-NYC.

Initially, the models using coarser, shared embeddings (Levels 2 & 3) adapt more quickly, outperforming the POI-only model (Level 0). After approximately 1,500 trajectories, the higher-specificity Level 0 model achieves better peak accuracy. However, it also exhibits greater sensitivity to distribution shifts (e.g., dips around 4k and 6k samples). The full MiLES model successfully combines the stability of the high-level embeddings with the peak performance of the fine-grained ones.

### B.3 ABLATION OF GRID GEOMETRY

To validate our choice of a hexagonal grid over a more conventional square grid, we re-ran our evaluations of MiLES using a square grid tiling while keeping all other hyperparameters identical to the main experiments.

As shown in Table 9, the hexagonal grid model consistently outperformed its square-grid counterpart. The performance gap was particularly pronounced on the GeoLife dataset, which does not feature POI information and therefore relies solely on grid-based embeddings. While specifically tuning the hyperparameters of MiLES for a square grid would likely yield improvements, these results empirically support the theoretical advantages of hexagonal grids for better representing spatial proximity in mobility tasks (Ke et al., 2019).

Table 9: Performance comparison of MiLES with hexagonal vs. square grids, using the BiTULER backbone. Results show top-1 accuracy, top-5 accuracy and macro F1. Best results in bold.

| Dataset | Grid Shape | Acc@1 | Acc@5 | F1 |
|---|---|---|---|---|
| Foursquare-NYC | Hexagon | **61.61** | **70.78** | **59.54** |
| | Square | 61.50 | 70.70 | 59.41 |
| Foursquare-TKY | Hexagon | **62.72** | **75.95** | **60.40** |
| | Square | 62.58 | 75.92 | 60.26 |
| GeoLife | Hexagon | **46.28** | **77.83** | **32.96** |
| | Square | 37.01 | 72.82 | 25.39 |

### B.4 AGGREGATION OF EMBEDDING LEVELS

To validate our choice of concatenation for aggregating MiLES's embedding levels, we compared it against summation. Figure 5 shows the results on the 399-user Foursquare-NYC dataset using BiTULER with fixed level weights.

When summing embeddings, adding more levels degrades top-2 accuracy, suggesting that the less-precise, high-level embeddings interfere with the fine-grained POI embeddings crucial for distinguishing individual users. While summation improves top-5 accuracy in some cases (as coarser features can help identify user groups), the concatenation-based approach consistently achieves the highest performance for both metrics. This empirically supports our design choice for MiLES.

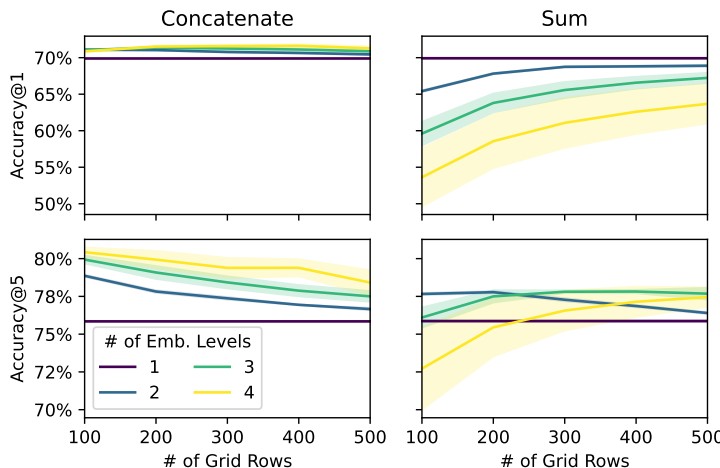

Figure 5: Performance on the 399 user Foursquare-NYC variant, depending on the grid resolution, the number of embedding levels and whether level-specific embeddings were concatenated or summed. BiTULER was used as the classifier and attention weights $w_l$ were fixed. Shaded areas represent the $1\sigma$ range.

## C  ADDITIONAL EXPERIMENTS AND BROADER CONTEXT

### C.1  APPLICATION OF MiLES TO DESTINATION PREDICTION

To demonstrate the general applicability of MiLES beyond Trajectory-User Linking, we conducted additional experiments on the task of destination prediction. This task aims to predict the final location of a trajectory given an initial partial sequence of its check-ins.

**Experimental Setup**   Our experimental setup is based on the ECML/PKDD 2015 discovery challenge on taxi destination prediction de Brébisson et al. (2015). For each dataset, we generated a stream of 50,000 pairs of partial trajectories and their final destinations by sampling sub-trajectories at random points. For the GeoLife dataset, we first filtered out check-ins outside the greater Beijing area and subsampled the remaining data at 10-minute intervals.

We framed the task as a regression problem, using a projection of the destination coordinates $l_n = [\text{longitude}_n, \text{latitude}_n]$ as labels. This enabled us to calculate the straight line distance between the actual- and predicted coordinates $\hat{l}_n$ as $||\hat{l}_n - l_n||_2$.

In terms of model architecture, we also employed a similar approach to that used in de Brébisson et al. (2015): We encode partial trajectories, as well as the weekday and user-ID associated with the respective trajectory. For the latter, we use embedding vectors with 256 dimensions each. All encoded features are then fed to a fully connected layer, producing the final prediction.

Previous work used a variety of neural architectures like multi-layer perceptrons (MLPs) (de Brébisson et al., 2015), convolutional neural networks (CNNs) (Lv et al., 2018) or LSTMs (Endo et al., 2017; Ebel et al., 2020) for the purpose of destination prediction. Based on this, we used either a bidirectional LSTM (BiLSTM), a convolutional neural network (CNN), a transformer (Transformer) or an MLP to encode the partial trajectories For the latter, we computed the average of all check-in embeddings to aggregate the partial trajectory.

We tune the hyperparameters of all models and techniques using the initial 5000 partial trajectories of the 400 user Foursquare-TKY dataset

**Results**   We evaluated all models using either a default single-level POI lookup embedding or the proposed MiLES embedding. Table 10 presents the mean, median (P50), and 90th percentile (P90) of the prediction errors, averaged over five prequential evaluation runs.

Table 10: Mean, 50th-, and 90th percentile of distances between predicted and actual destinations in kilometers for models using a basic lookup embedding (Default) and performance gains when using our technique ($\Delta$ MiLES).

| Dataset | Foursquare-NYC | | | Foursquare-TKY | | | GeoLife | | |
|---|---|---|---|---|---|---|---|---|---|
| Model | Mean | P50 | P90 | Mean | P50 | P90 | Mean | P50 | P90 |
| Default | | | | | | | | | |
| BiLSTM | 7.21 | 5.37 | 20.03 | 6.97 | 5.60 | 17.68 | **5.96** | **4.29** | **16.90** |
| CNN | **6.51** | **4.81** | **18.28** | **6.57** | **5.21** | **16.86** | 6.33 | 4.56 | 17.87 |
| MLP | 7.80 | 6.14 | 20.08 | 7.53 | 6.23 | 18.09 | 7.12 | 5.39 | 19.10 |
| Transformer | 8.90 | 7.26 | 21.81 | 8.11 | 6.93 | 18.58 | 7.54 | 5.81 | 19.97 |
| $\Delta$ MiLES | | | | | | | | | |
| BiLSTM | -0.72 | -0.65 | -1.56 | -0.24 | -0.29 | -0.24 | -0.09 | -0.07 | -0.34 |
| CNN | -0.13 | -0.18 | -0.11 | 0.05 | 0.04 | 0.13 | -0.20 | -0.20 | -0.28 |
| MLP | -0.55 | -0.52 | -0.99 | -0.16 | -0.18 | -0.21 | -0.16 | -0.17 | -0.25 |
| Transformer | -1.01 | -0.97 | -1.89 | -0.48 | -0.51 | -0.59 | -0.09 | -0.09 | -0.17 |

Table 11: Mean, 50th-, and 90th percentile of distances between predicted and actual destinations in kilometers for different embedding- and experience replay methods, averaged across all models shown in Table 10. [†]The default configuration uses POI-based lookup embeddings without replay.

| Dataset | Foursquare-NYC | | | Foursquare-TKY | | | GeoLife | | |
|---|---|---|---|---|---|---|---|---|---|
| Method | Mean | P50 | P90 | Mean | P50 | P90 | Mean | P50 | P90 |
| Default[†] | 7.61 | 5.90 | 20.05 | 7.47 | 6.14 | 18.22 | 6.74 | 5.01 | 18.46 |
| Embedding | | | | | | | | | |
| Lookup | 7.61 | 5.90 | 20.05 | 7.30 | 5.99 | 17.80 | 6.74 | 5.01 | 18.46 |
| Linear | 7.13 | 5.56 | **18.63** | 7.20 | 5.92 | **17.55** | 6.83 | 5.12 | 18.53 |
| Fourier | 7.00 | 5.34 | 18.88 | 7.14 | 5.80 | 17.73 | 6.69 | 4.96 | 18.41 |
| MiLES | 7.00 | 5.32 | 18.91 | **7.09** | 5.76 | 17.57 | 6.60 | 4.88 | 18.20 |
| Replay | | | | | | | | | |
| FIFO | 7.39 | 5.52 | 20.35 | 7.30 | 5.82 | 18.64 | 6.37 | 4.50 | 18.33 |
| Random | 7.39 | 5.50 | 20.41 | 7.31 | 5.82 | 18.62 | 6.31 | 4.43 | 18.27 |
| FIFO+MiLES | **6.82** | **4.97** | 19.35 | 7.10 | **5.56** | 18.45 | **6.09** | **4.26** | **17.67** |

Among the baseline models, the CNN performed best on the Foursquare datasets, while the BiLSTM was superior on GeoLife. The Transformer model yielded the worst results across all datasets, which is likely attributable to the short average length of the partial trajectories (e.g., only five check-ins for Foursquare-NYC), limiting the effectiveness of its self-attention mechanism.

Consistent with our TUL experiments, integrating MiLES improved destination prediction performance across all models, metrics, and datasets. The Foursquare datasets, which have fewer check-ins per trajectory, benefited most, with MiLES achieving reductions in mean error of up to 11%. The performance gains were less pronounced for the CNN model. We hypothesize that this is because the local connectivity of convolutional kernels already causes some similarity between embeddings of nearby POIs, an effect that partially overlaps with the explicit spatial sharing in MiLES. We further compared MiLES against alternative embedding and experience replay techniques, with results averaged over all models reported in Table 11. While linear embeddings achieved a lower 90th percentile error on the Foursquare datasets, MiLES consistently demonstrated the best performance in terms of mean and median prediction error compared to the default lookup and other hybrid embeddings.

Experience replay strategies significantly reduced mean and median errors. However, they did not improve the 90th percentile error, suggesting that while replay reinforces common travel patterns, it may not help with predicting less frequent or novel destinations that constitute the long-tail of the error distribution. Notably, the combination of MiLES with a FIFO replay buffer (FIFO + MiLES)

achieved the best results across nearly all metrics and datasets, once again demonstrating that MiLES can be used to complement and enhance other online learning strategies.

## C.2 CONCEPT DRIFT IN TRAJECTORY DATA

As mentioned above, real-world applications based on mobility data are likely to face changes in the data distribution in the form of concept drift. Such drifts can also be found in the datasets used in this paper. Figure 6 provides a visual example of concept drift found in the GeoLife dataset. While the trajectories recorded earlier in the collection period and shown in a darker hue are relatively

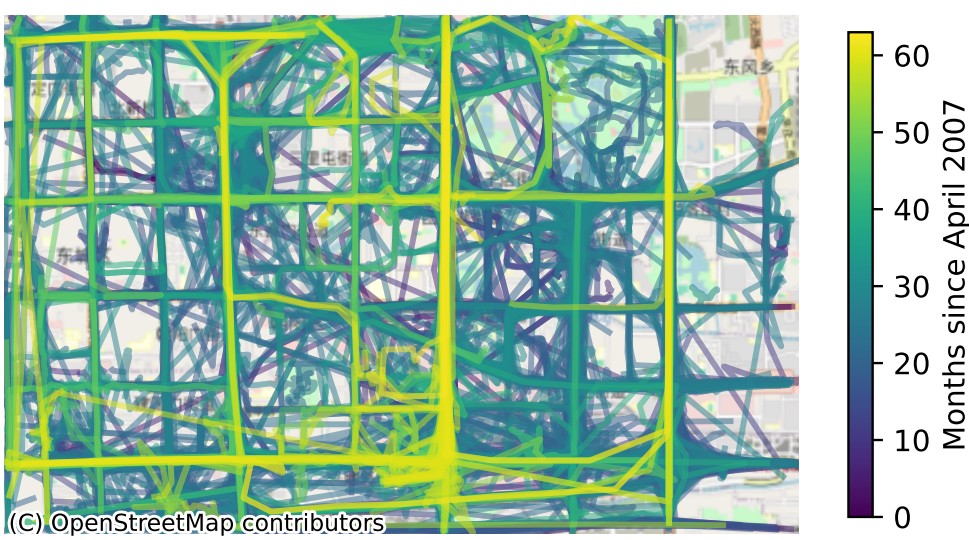

Figure 6: Distribution of trajectories in the GeoLife dataset over time for highly frequented area. Trajectories are color-coded according based on their time of occurrence.

evenly distributed across the area, the later trajectories are much more concentrated around the major transportation axes. This effect could be caused by a change in either user behavior or data collection methodology.

## C.3 PRIVACY CONSIDERATIONS

The primary focus of this work is on the technical performance of Trajectory-User Linking (TUL) in online learning settings. However, we acknowledge the significant privacy implications inherent in any system that processes user-specific mobility data. TUL applications handle sensitive information about individual movements, which could be misused if not properly protected.

While the fundamental privacy challenges of TUL are consistent between batch and online settings, the continuous nature of online learning may introduce distinct risks. For instance, per-instance model updates could potentially allow an adversary to infer changes in an individual's routine more rapidly than with periodic batch updates. The core privacy issues are extensively discussed in prior work (Vincent et al., 2019; Jiang et al., 2021), with common mitigation strategies including data obfuscation and differential privacy (e.g. Mir et al., 2013; Yin et al., 2018).

