# OpenReview forum: "Multi-Level Spatial Embedding Sharing for Enhanced Online Trajectory-User Linking"
_ICLR.cc/2026/Conference — ICLR 2026 Conference Withdrawn Submission_

### Official Review · Reviewer_TTJB · 2025-10-23

**Soundness:** 3
**Presentation:** 3
**Contribution:** 1
**Rating:** 2
**Confidence:** 4

**Summary:**

This paper is a pioneering work on the trajectory-user linking task under the online learning context. The paper identifies that the primary challenge of configuring existing TUL methods with the online learning setting is the embedding of POIs, where a too fine-grained embedding will face sparsity challenges, while a too coarse-grained embedding will lose expressivity. The paper then proposes a solution where multiple granularities of embeddings are concatenated together to mitigate the problem. Experiments show that existing methods see a performance improvement after applying the proposed embedding solution.

**Strengths:**

1. The paper represents a pioneer work to explore the trajectory-user linking task under the online learning context, which is relevant given the real-world use cases of TUL.
2. The presentation of the paper is straightforward and intuitive. The presentation of the motivation and design choices is detailed and easy to understand.
3. Experiments show that the proposed embedding strategy can improve the performance of various existing TUL methods under online learning context.

**Weaknesses:**

The biggest weakness of this paper is the lack of technical contribution. The proposed solution essentially follows the core idea of combining multi-level embeddings, which is quite common in the context of trajectory and POI check-in data mining, given that it is how Geohash is constructed and is also a technique used in many existing methods. It doesn't seem that the paper introduced any ground-breaking novelty on top of that core idea.

**Questions:**

Could the authors elaborate on the novelty and technical significance of their proposed solution?

---

### Official Review · Reviewer_juRx · 2025-10-25

**Soundness:** 3
**Presentation:** 3
**Contribution:** 2
**Rating:** 4
**Confidence:** 4

**Summary:**

This paper introduces a multi-level spatial embedding framework for the trajectory user linking problem, which combines locations encoded and shared across multiple spatial levels. Experiments using various TUL methods with their embedding techniques replaced with MiLSE on Foursquare datasets NYC and TKY, and GeoLife, demonstrate that multi-level embedding is helpful in trajectory representation learning. The method also demonstrates broad applicability, with improvements shown in the separate task of destination prediction.

**Strengths:**

+ MiLSE proposes a simple solution to the core challenge of gradient sparsity in location embeddings for online TUL.

+ Good ablation studies confirm the importance of the multi-level design. Ablation settings confirm the effective feature learning across multiple levels.

+ The task is relevant to real-world mobility applications dealing with continuous data streams and concept drift.

+ The link to the code repo is provided.

**Weaknesses:**

- The ablations in Fig. 2(a) show that without L1, the performance is roughly the same. Fig 2(b) also shows that the performance gain from 1 – 4 levels is minor. This undermines the efficacy of multi-level embedding.
-  The multi-level embedding introduces a minor improvement on the NYC dataset and minor computational efficiency.
- This seems an incremental improvement based on existing TUL methods, and there are also limited experiments/tasks to demonstrate that MiLSE is effective in encoding trajectory features.

**Questions:**

Additional question: could you please explain why multi-level hexagons sometimes are more efficient than the default embeddings, and less parameters?

**Details Of Ethics Concerns:**

All datasets are anonymous, no ethics concerns.

---

### Official Review · Reviewer_ihUy · 2025-10-27

**Soundness:** 3
**Presentation:** 2
**Contribution:** 2
**Rating:** 2
**Confidence:** 5

**Summary:**

The paper proposes a framework for trajectory user linking (TUL) embedding in an online learning context. The main idea is to define a hierarchical embedding space that allows partial sharing of embeddings for locations within neighbourhoods. The authors have evaluated the method's performance across three benchmark datasets. Several trajectory-user linking methods are implemented within this framework.

**Strengths:**

- The strength of the method is proposing a general framework for online learning that can be used in combination with many TUL approaches and potentially in other tasks.
- Hyperparameter tuning is performed for all methods during evaluation
- For evaluation, several TUL methods are implemented under the framework

**Weaknesses:**

- Statistical significance tests are missing; the results seem to be from one run. Please perform multiple runs to account for randomness.
-  While concept drift adaptation is mentioned as motivation in the introduction. This aspect is not properly evaluated. Only in Appendix C. 2, the existence of a trajectory with concept drift is presented. The effectiveness of the method against concept drift, for instance, can be evaluated using simulated data under different proportions of concept drift.
- In general, the paper can benefit from a better writing flow. Please maintain a balance between paragraph length (too short and too long) and use topic sentences for each paragraph. The contributions of the paper are not explicitly mentioned in the introduction. From the related work section, it becomes clear that the concept of multi-granularity has been proposed before. It is important that the introduction makes the paper's novel aspects completely clear. TUL: The methods used to implement the framework are not introduced or motivated in the experiments section (the method, learnable linear projection, lacks a reference).
- The methodology section only provides the logic and intuition behind the method, but algorithmic details are not provided. Given that this is an online learning algorithm, it should work under an initially unknown number of users, locations, geographic areas, etc. However, it seems to me that the number of locations and the grid area are already clear. It is not clear which parameters should be known beforehand, and if the method can really work in an online setting, starting with no information on the dynamic number of users and POIs in changing geographic regions.

- line 176 -> d is not introduced

**Questions:**

- In Table 3, on which basis do you consider the comparison between methods to be fair, e.g., do all methods have the same embedding length?
- Can you clarify the minimum input needed for the algorithm to run in an online setting?

---

### Official Review · Reviewer_orpQ · 2025-10-29

**Soundness:** 3
**Presentation:** 4
**Contribution:** 3
**Rating:** 8
**Confidence:** 4

**Summary:**

The research addresses what is described as a fundamental challenge in Trajectory-User Linking (TUL), which involves identifying people based on their movement patterns throughout the day. Traditional machine learning models struggle with streaming location data because they require all data upfront and cannot adapt quickly when data arrives continuously or when behavioral patterns shift over time. The core technical problem lies in how locations are represented: conventional approaches use individual embeddings for each location, but since most places are rarely visited, the models learn slowly and struggle to adapt to new patterns or changing behaviors.

The proposed solution, MiLES (Multi-Level Spatial Embedding Sharing), represents locations at multiple hierarchical scales simultaneously—from specific individual locations to neighborhoods, districts, and broad regions. This multi-scale approach enables the model to learn efficiently by sharing information across spatially related locations, allowing it to quickly adapt when observing one location while maintaining fine-grained details about specific places.

This paper claims up to 24% improvement in user identification accuracy while using less computational resources, with practical applications spanning public health tracking, ride-sharing optimization, urban planning, and location-based recommendation systems, when tested on real-world datasets collected from FourSquare.

**Strengths:**

I found this a well-written, technically-sound paper that addressed an interesting problem. The improvement in accuracy (24%) is significant.

**Weaknesses:**

Improvement in computational speed is claimed as an added contribution; however this improvement appears to be marginal.

**Questions:**

N/A

---

### Note · Authors · 2025-11-13

**Comment:**

Dear reviewers,
We appreciate your time and constructive feedback on our submission. We have decided to withdraw the manuscript to address the reviewers’ comments and prepare a revised version for a potential resubmission.

**Withdrawal Confirmation:**

I have read and agree with the venue's withdrawal policy on behalf of myself and my co-authors.